# Dynamical barrier and isotope effects in the simplest substitution reaction via Walden inversion mechanism

Zhiqiang Zhao[1,2,*], Zhaojun Zhang[1,*], Shu Liu[1] & Dong H. Zhang[1,3]

Reactions occurring at a carbon atom through the Walden inversion mechanism are one of the most important and useful classes of reactions in chemistry. Here we report an accurate theoretical study of the simplest reaction of that type: the $H + CH_4$ substitution reaction and its isotope analogues. It is found that the reaction threshold versus collision energy is considerably higher than the barrier height. The reaction exhibits a strong normal secondary isotope effect on the cross-sections measured above the reaction threshold, and a small but reverse secondary kinetic isotope effect at room temperature. Detailed analysis reveals that the reaction proceeds along a path with a higher barrier height instead of the minimum-energy path because the umbrella angle of the non-reacting methyl group cannot change synchronously with the other reaction coordinates during the reaction due to insufficient energy transfer from the translational motion to the umbrella mode.

[1] State Key Laboratory of Molecular Reaction Dynamics, Dalian Institute of Chemical Physics, Chinese Academy of Sciences, Dalian 116023, Liaoning, China. [2] University of Chinese Academy of Sciences, Beijing 100049, China. [3] Center for Advanced Chemical Physics and 2011 Frontier Center for Quantum Science and Technology, University of Science and Technology of China, Hefei 230026, China. * These authors contributed equally to this work. Correspondence and requests for materials should be addressed to S.L. (email: liushu1985@dicp.ac.cn) or to D.H.Z. (email: zhangdh@dicp.ac.cn).

Reactions occurring at a carbon atom in a tetrahedral environment, proceeding through the back-side attack Walden inversion mechanism leading to inversion of the chirality of a molecule, are one of the most important and useful classes of reactions in chemistry. The bimolecular nucleophilic substitution ($S_N2$) reaction is the most common reaction of that type, in which a nucleophile (often negatively charged) approaches a saturated carbon from one side, displaces a leaving group on the opposite side of the carbon atom, resulting in inversion of the carbon centre. Strong solvent effects of these reactions in solution have prompted investigations of the gas-phase $S_N2$ reaction to probe the intrinsic reaction mechanisms without solvent[1–27]. Numerous experimental and theoretical studies have revealed that an inverse secondary kinetic isotope effect (KIE), that is, $k_H/k_D < 1$ (where $k$ denotes the thermal rate constant), is characteristic for a thermal $S_N2$ reaction when the isotopically substituted atom is not directly involved in a reaction[1–10]. In strong contrast, by using a guided ion beam technique, Ervin and co-workers discovered that the symmetric $^{37}Cl^- + CH_3^{35}Cl \rightarrow ^{35}Cl^- + CH_3^{37}Cl$ reaction exhibits a large normal secondary isotope effect on the cross-section, in addition to a reaction threshold versus collision energy substantially higher than the calculated barrier height[11,14]. They also observed that the reaction threshold for the endothermic $Cl^- + CH_3F \rightarrow CH_3Cl + F^-$ reaction is considerably higher than the reaction endothermicity[12]. Hase and co-workers performed extensive direct molecular dynamics studies on many exothermic and a few centre barrier $S_N2$ reactions, and uncovered the non-statistical nature and inefficiency of energy transfer between intermolecular and intramolecular modes in these reactions[13–18]. Furthermore, they found that the reaction thresholds for $S_N2$ reactions, which are larger than their corresponding barrier heights, result from direct reactions[15,17]. Many reduced dimensionality quantum scattering studies were also carried out to investigate dynamics in $S_N2$ reactions[19–22]. Recently, the molecular beam experiments in the Wester's group, in combination with theory, have revealed unprecedented dynamical details on some exothermic $S_N2$ reactions[23–27]. However, despite so many experimental and theoretical investigations, the origin of the intriguing difference between the isotope effects on the kinetic and cross-sections as well as the high-energy threshold for centre barrier and endothermic $S_N2$ reactions remain unclear.

The $H + CH_4$ reaction is the simplest and most prototypical reaction occurring at a carbon atom in a tetrahedral environment. The abstraction $H + CH_4 \rightarrow H_2 + CH_3$ reaction has been the subject of numerous experimental investigations, and has also become a benchmark system for theoretical studies of polyatomic reactions[28–39]. In addition to the abstraction reaction, there exists a substitution reaction, $H' + CH_4 \rightarrow H + CH'H_3$, with a $D_{3h}$ transition state and a static barrier height of 1.6 eV. It is the simplest reaction proceeding through the back-side attack Walden inversion mechanism, very similar to the gas-phase $S_N2$ reactions with central barriers, except that there exist pre- and post-reaction wells in $S_N2$ reactions stemming from strong ion–dipole interaction between reagents/products. The experimental study of the substitution reaction has a long history. Rowland and co-workers observed a threshold energy of $\sim 1.5$ eV for the $T + CD_4 \rightarrow CD_3T + H$ reaction[40], and obtained a relative yield of 7.2 for the $T + CH_4$ and $T + CD_4$ reaction using tritium atoms with a translational energy of 2.8 eV (ref. 41). Bersohn and co-worker measured the substitution cross-sections for the $H + CD_4 \rightarrow CHD_3 + D$ and $H + CH_3D \rightarrow CH_4 + D$ reactions at a collision energy near 2 eV to directly study the secondary isotope effects, and found that the cross-section was reduced by about a factor of 2 from $0.040 \pm 0.015\,\text{Å}^2$ for $H + CH_3D$ to $0.021 \pm 0.005\,\text{Å}^2$ for $H + CD_4$ (per D atom), indicating that the

secondary isotope effect is strong in the reaction, although not as prominent as observed by Rowland and co-workers[42]. On the basis of measured cross-sections and the velocity distribution measured for the reacting H atom and displaced D atom, they concluded that the substitution reaction takes place by an inversion mechanism.

Despite these intriguing experimental discoveries, to the best of our knowledge there is no theoretical study of the substitution reaction on a high-quality potential energy surface (PES), except Bunker and co-workers performed trajectory calculations on a model analytic PES[43]. Quantitative theory for such a reaction faces two difficult tasks: the construction of an accurate global PES involving six atoms and the performance of reactive scattering calculations, preferably quantum mechanical, on an accurate PES. The past decade has witnessed significant progress on accurate PES construction and quantum reactive scattering study of polyatomic reactions[38,39,44–50]. These advances have been combined with the rise in computational power to make accurate dynamics study practical for such a reaction process.

Here we report an accurate theoretical study of the $H + CH_4$ substitution reaction. Our calculation shows the reaction has (a) a threshold energy considerably higher than the static barrier height, (b) a large normal secondary isotope effect on cross-section and (c) a small but reverse secondary KIE, in exactly the same way as have been observed in many $S_N2$ reactions. Therefore, the study not only provides unprecedented dynamical details for this simplest Walden inversion reaction, but also sheds light on the dynamics of gas-phase $S_N2$ reactions.

## Results

**Reaction threshold energies and dynamical barrier heights.** The total reaction probabilities for the substitution reactions

$$H' + CH_4 \rightarrow H + CH_3H' \qquad (1)$$

$$H' + CH_3D \rightarrow D + CH_3H' \qquad (2)$$

$$H' + CHD_3 \rightarrow H + CD_3H' \qquad (3)$$

are presented as a function of collision energy for their corresponding ground rovibrational initial states in Fig. 1. (Note the probability for reaction (1) is for one H atom substitution). The

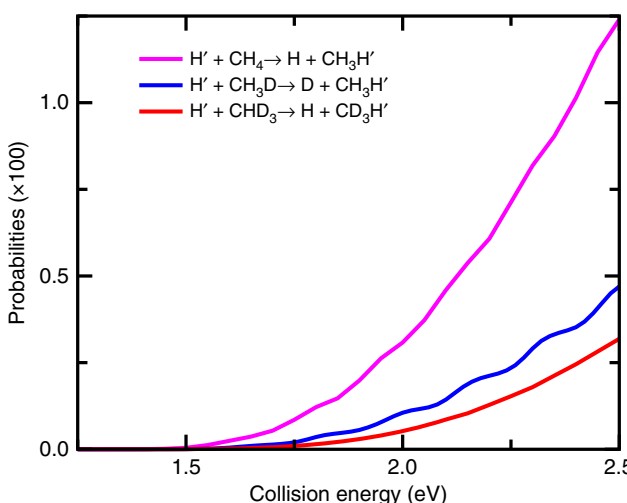

**Figure 1 | Reaction probabilities for the ground rovibrational initial states.** Total reaction probabilities for the total angular momentum $J = 0$ for reactions (1–3) involving ground-state reactants as a function of collision energy for the ground rovibrational initial state.

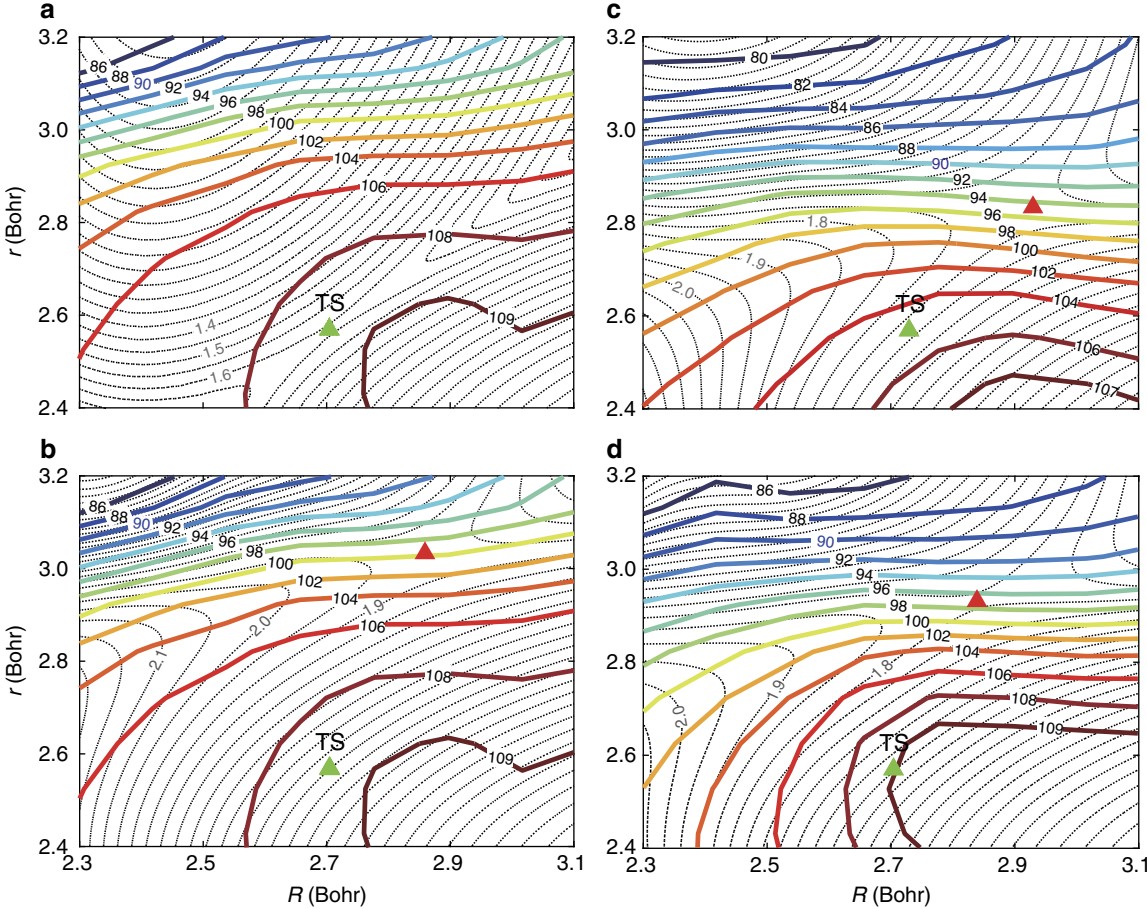

**Figure 2 | Dynamical barrier analyses.** (**a**) The average value of the umbrella angle $<\chi>$ for methyl for reaction (3) calculated from the scattering wavefunction at the translational energy $E = 1.8$ eV as a function of $R$ (the distance between incoming H and the centre of mass of CHD$_3$) and $r$ (the bond length of breaking CH bond) for the ground initial state (shown in colour contour lines with corresponding values indicated), together with the potential energy contour obtained by minimizing the other degrees of freedom (shown in grey contour lines). The static saddle point is marked by the green triangle; (**b**) same as **a** except the potential contour calculated by taking the umbrella angle at the calculated average value for every combination of $R$ and $r$. The dynamical saddle point is marked by the red triangle; (**c**) same as **b** except for reaction (1); (**d**) same as **b** except for the first umbrella excited initial state.

reaction probabilities increase smoothly and quickly with the increasing of the translational energy, but the overall values for all the three reactions are small. The probability for reaction (1), which is the largest among these three reactions, only reaches 0.0124 at $E = 2.5$ eV. With the substituted atom changed from H to D, the substitution probability for reaction (2) drops substantially, by a factor of 2.6 at $E = 2.5$ eV. This large primary isotope effect, as also found in the H′ + H$_2$O → H′OH + H and H′ + HOD → H′OH + D reactions[51] is apparently due to the fact that the substituted H atom is faster than D atom in responding to the attack of the incoming H atom.

In contrast to the H′ + H$_2$O → H′OH + H and H′ + HOD → H′OD + H reactions[51], the reaction probability for reactions (1) and (3), with the same newly formed bond and cleaved bond, decreases substantially as the non-reacting group changes from CH$_3$ to CD$_3$, manifesting strong secondary isotope effects. As these two reactions process on the same PES with their variational bottlenecks located at the saddle point and their zero point energies (ZPEs) corrected barrier heights differ < 0.01 eV (1.591 versus 1.585 for reactions (1) and (3)), the substantial difference between two reactions apparently cannot be explained by the barrier height or the topography of the PES.

To explore the dynamics difference between these two isotopic reactions, we calculated the average value of the umbrella angle $\chi$

for methyl from the scattering wavefunction at the translational energy $E = 1.8$ eV. Figure 2a shows the average value of the umbrella angle, $<\chi>$, as a function of $R$ (the distance between incoming H and the centre of mass of CHD$_3$) and $r$ (the bond length of breaking CH bond) for the H + CHD$_3$ substitution reaction, together with the potential energy contour obtained by minimizing the other degrees of freedom. The static saddle point locates at $R = 2.7$ Bohr and $r = 2.5$ Bohr with the corresponding umbrella angle $\chi = 90°$. However, the calculated $<\chi>$ equals to $108.5°$ at the static saddle point, close to the corresponding vibrationally averaged value for CHD$_3$ reagent of $109.36°$. With the increase of $r$ (elongation of CH bond) or as the wavefunction proceeding to the product region from the static saddle point, $<\chi>$ decreases in an accelerated way, eventually passes through the $90°$ line at $r \sim 3.1$ Bohr, which is considerably larger than the value at the saddle point. This means that the reaction does not proceed along the minimum-energy path on which the umbrella angle $<\chi>$ changes synchronously during the reaction with the incoming of H atom and elongation of the breaking CH bond to reach a value close to $90°$ at the static saddle point. Instead, it essentially does not vary while the incoming H atom approaches to the C atom and the breaking bond CH bond is elongated to their corresponding desired values for reaction. Only after that, the umbrella angle $\chi$ starts to decrease, initially slowly, then

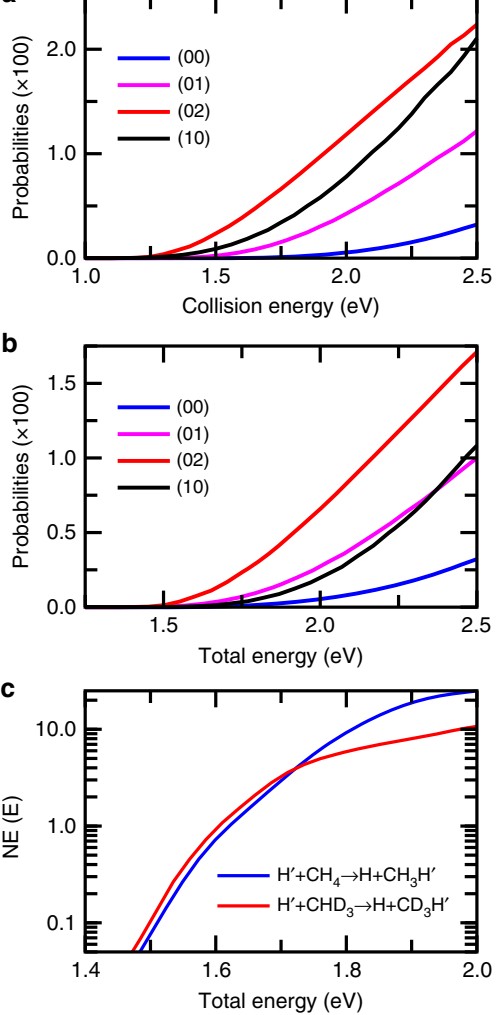

**Figure 3 | Reaction probabilities for vibrational excited states and cumulative reaction probabilities.** (**a**) Total reaction probabilities for a number of initial vibrational states for reaction (3) as a function of translational energy; (**b**) same as **a** except as a function of total energy measured from the ground-state energy of $CHD_3$; (**c**) the cumulative reaction probabilities for reactions (1) and (3) as a function of total energy.

increasingly fast. Therefore, in reality, the reaction proceeds not on the potential shown in Fig. 2a, but on a potential shown in Fig. 2b, which is obtained by taking the umbrella angle at the calculated average value for every combination of $R$ and $r$. In Fig. 2b, the 'saddle point' moves to $R \sim 2.85$, $r \sim 3.1$ Bohr. This is the very dynamical saddle point for the reaction with a corresponding barrier height of $\sim 1.93$ eV, higher than the original barrier height by 0.32 eV, also locating much later than the original barrier in term of C–H bond length.

The above finding clearly reveals that the umbrella motion of the non-reacting $CD_3$ group is slow in responding to the attack of the incoming H atom during the reaction. Because the umbrella motion depends on the moment of inertia (masses of atom in the methyl group), it is expected that isotope replacement of D atom to H atom can change the umbrella motion substantially. As seen from Fig. 2c for the $H + CH_4$ reaction, the umbrella angle starts to decrease even before the incoming H atom reaches the static saddle point, reaches 90° at $r \sim 2.9$ Bohr in an accelerated way, and then begins to slow down with the further increase of $r$. As a result, the reaction has a dynamical saddle point at $r \sim 2.8$ Bohr

with a corresponding dynamical barrier height of 1.73 eV, lower than that for $H + CHD_3$ by 0.2 eV. Therefore, it is clear that the strong secondary isotope effects in the reaction system in term of the huge differences on the reaction thresholds versus collision energy and the magnitudes of reaction probabilities for these two isotopic reactions shown in Fig. 1 originate from the important effect of the umbrella motion of the methyl group on the reactions.

**Effects of reagent vibrations on the reaction.** Figure 3a compares the reaction probabilities for a number of initial vibrational states for reaction (3) as a function of collision energy. Both the CH stretch excitation and $CD_3$ umbrella excitation substantially enhance the reactivity and reduce the reaction threshold versus collision energy (see Supplementary Fig. 2 for more initial states and Supplementary Fig. 3 for reaction (1)). The reaction probability for the first umbrella excited (0,1) state, with an excitation energy of 0.126 eV, is higher than that for the ground state by a factor of 67, 8.0 and 3.8, respectively, at $E = 1.5$, 2.0 and 2.5 eV. The reactivity for the (0,2) state, with an excitation energy of 0.252 eV, is even higher than that for the first CH stretch excited state with an excitation energy of 0.369 eV, in particular in low energy region, indicating that the umbrella excitation is more efficient on promoting the reaction than CH stretch excitation. This can be seen more clearly from Fig. 3b, which shows these reaction probabilities as a function of total energy measured from the ground vibrational of $CHD_3$ (see Supplementary Fig. 4 for more initial states and Supplementary Fig. 5 for reaction (1)). Both the CH stretch excitation and $CD_3$ umbrella excitation are more efficient than the translational energy on promoting the reaction, in particular for the umbrella motion the efficacy increases with initial excitation up to the 5th/6th overtone state (Supplementary Fig. 4). For reaction (1), the situation is similar as for reaction (3), with both the CH stretch excitation and $CH_3$ umbrella excitation more efficient than the translational energy on promoting the reaction (Supplementary Fig. 5).

Because reaction (3) possess a late barrier, in particular a very late dynamical barrier, in term of the CH stretch as shown in Fig. 2a,b, it is expected that the CH stretch excitation enhances the reactivity substantially, according to Polanyi's rules, which state that vibrational energy is more efficient than translational energy in promoting a late barrier reaction[52]. To understand the pronounced effect of the umbrella excitation on the reactivity, we present in Fig. 2d the average umbrella angle for methyl group as a function of $R$ and $r$ for the $H + CHD_3(0,1)$ reaction at the translational energy $E = 1.8$ eV. Around the static saddle point, the average umbrella angle for the (0,1) state is very close to ground state shown in Fig. 2a. However, with the further elongation of $r$, the umbrella angle for the excited state decreases much faster than the ground state, resulting in a dynamical barrier height of $\sim 1.80$ eV at $(R = 2.9, r = 2.9)$. This dynamical barrier is lower than the ground state by $\sim 0.19$ eV, larger than the (0,1) excitation energy of 0.126 eV. Clearly, more vibrational energy in the umbrella mode due to initial excitation makes the umbrella angle faster to respond, giving rise to a reduced dynamical barrier height and enhancing the reactivity substantially. Therefore, the large difference on reactivity between the $H + CH_4/CHD_3$ reactions shown in Fig. 1 should also relate to the difference in ZPE, in addition to the difference in the moment of inertia discussed above. This, on the other hand, also indicates that the energy transfer from translational motion to umbrella mode is inefficient because an efficient energy transfer to the umbrella mode would enhance the efficacy of translational energy on prompting reaction.

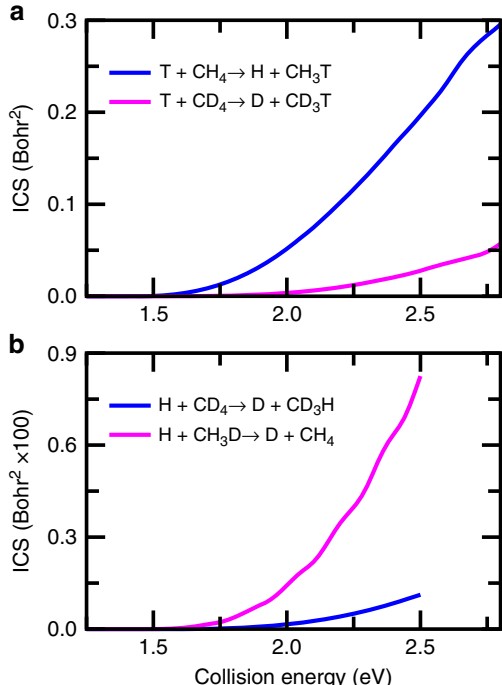

**Figure 4 | Integral cross-sections.** (**a**) The integral cross-sections for the $T + CH_4/CD_4$ substitution reactions as a function of collision energy; (**b**) same as **a** except for the $H + CD_4/CH_3D \rightarrow D + CD_3H/CH_4$ reactions.

From Fig. 3a, one can see that for the $H + CHD_3$ reaction, the reactivity at a given collision energy above threshold, in particular at high energy, is mainly determined by that for the ground vibrational state, because even the first umbrella excited state with an excitation energy of 0.126 eV has a small population at a moderate temperature (Say, at $T = 300$ K the population is $\sim 0.75\%$). This means that measured at a given collision energy, the cross-sections for reaction (1) will be substantially larger than that for reaction (3), that is, one will observe a large normal secondary isotope effect on the integral cross-section.

However, the thermal rate constant for the reaction, as can be seen from Fig. 3b, is contributed by the vibrational excited states, in particular the highly umbrella excited states (Supplementary Fig. 4). With many vibrationally excited states contributed to the thermal rate constant, it is more efficient to calculate the cumulative reaction probabilities, NE(E), (the sum of the reaction probabilities for all the initial state with a fixed total energy) for the reactions (1) and (3) from which the thermal rate constants can be reliably estimated[35,39] (see Supplementary Information for details). Figure 3c shows the NE(E) for reactions (1) and (3) as a function of total energy measured from their corresponding ground-state energies. Surprisingly, although the reaction probabilities for reaction (3) for the ground initial state is substantially smaller than reaction (1) as shown in Fig. 1, NE(E) for the reaction (3) is actually slightly larger than reaction (1) for the total energy up to 1.72 eV. On the basis of NE(E), we obtained a thermal rate constant at 300 K for reaction (1) of $3.1 \times 10^{-36}$ cm$^3$ s$^{-1}$, for reaction (3) of $3.5 \times 10^{-36}$ cm$^3$ s$^{-1}$, which results in $k_H/k_D = 0.89$, a reverse KIE. It is worthwhile to point out here that the tiny rate constant is due to the 1.62 eV static barrier height of the reaction, but the relative rate constant is still a good indicator of KIE.

The cumulative reaction probabilities shown in Fig. 3c for reactions (1) and (3) look intriguing because the reaction probability for the ground vibrational state for reaction (1) is

much larger than that for reaction (3) as shown in Fig. 1. From the initial state selected reaction probability point of view, larger NE(E) for reaction (3) indicates contributions from the umbrella excited states for the reaction eventually beat those for reaction (1). While from the transition state point of view, this means in low energy the umbrella motion both for $CH_3$ and $CD_3$ can follow the motion of H atom to product region without much reflection of the wavefunction (or re-crossing of trajectory in classical terminology). Because the reaction (3) has a lower ZPE corrected barrier height and a relatively higher density of the umbrella states, it has a slightly larger NE(E). However, with the increase of energy, the umbrella motion of $CD_3$ cannot act as fast as $CH_3$ on responding to the H atom motion any more, causing sever reflection of the wavefunction that results in a smaller value of NE(E).

**Integral cross-sections and comparison with experiments**. Finally, we depict integral cross-sections for $T + CH_4/CD_4$ substitution reactions in Fig. 4a and for the $H + CD_4/CH_3D$ reaction in Fig. 4b, which have been measured experimentally. Same as the reaction probabilities for $J = 0$ shown in Fig. 1, large but normal isotope effects can be observed for the reaction cross-sections. Agreement between theory and experiment on the integral cross-sections is both positive and negative. We obtained a relative yield of 6.5 for the $T + CH_4/CD_4$ substitution reactions for tritium atoms with a translational energy of 2.8 eV, in comparison with the experimental value of 7.2 (ref. 41), despite the fact that there are some uncertainties in the comparison due to partial thermalization of the hot atoms before reaction in the experiment as discussed by Raff and co-workers[53]. However, for the $H + CH_3D$ and $H + CD_4$ reactions, the relative theoretical yield at the collision energy of 2.2 eV is 8.4, in comparison with experimental value of 2.0 (ref. 42). Even worse, the absolute theoretical integral cross-section for the $H + CH_3D \rightarrow CH_4 + D$ reaction is only 0.001 Å$^2$ at that collision energy, smaller than the reported experimental value by a factor of 20 (ref. 42). Although the PES used in the present study is highly accurate, the theoretical cross-sections presented here are only for the ground rotational states, while the experimental results contain the contributions from all the rotational states populated in the experiment. Further studies should be performed to investigate the effects of initial rotational excitation of reagent on the reactions. In addition, the reduced dimensionality model used in this study by constraining the non-reacting methyl group to $C_{3v}$ symmetry[32,34] may also introduce some errors, despite the fact that the model is at a quantitative level of accuracy for the abstraction reaction[39].

## Discussion

We have carried out a detailed theoretical study of the dynamics of the $H + CH_4$ substitution reaction and its isotope analogues. Our calculation reveals that the $H + CH_4$ substitution reaction may manifest different isotope effects. In terms of the cross-section beyond the reaction threshold energy, it has a large normal secondary isotope effect because the measured cross-section at a moderate temperature is mainly contributed by the ground vibrational state. However, in terms of the thermal rate constant, it has a reverse secondary KIE due to contributions from the umbrella excited states.

All of the phenomena found from our calculation for the $H + CH_4$ substitution reactions due to disparity between the translational motion and umbrella motion and inefficient energy transfer between them, including higher threshold energy, a reverse secondary KIE at room temperature and a large normal isotope effect for reaction cross-section, have been observed for

gas-phase $S_N2$ reactions with barriers. Extensive direct molecular dynamics simulations have revealed similar inefficient energy transfer as observed here and a non-statistical nature in $S_N2$ reactions, despite the fact there exist pre- and post-barrier wells in gas-phase $S_N2$ reactions[13–18]. Inefficient energy transfer between intermolecular and intramolecular modes render the umbrella mode inert to approach of the nucleophile until the nucleophile is very close to the C atom (the distance between the nucleophile and C atom close to that at the static saddle point), making the pre-barrier well not useful on activating the umbrella mode. Therefore, we anticipate that the mechanisms uncovered from this study may play important roles in these gas-phase $S_N2$ reactions. More accurate dynamics studies should be carried out on accurate PESs for $S_N2$ reactions with barriers to provide a more definitive answer.

## Methods

**Potential energy surface.** The PES used in the calculation is the full dimensional global PES constructed in this group with neural network method based on ~46,000 UCCSD(T)-F12a/aug-cc-pVTZ *ab initio* energies. With a fitting error is 4 meV, measured in term of root-mean-square error, the PES is highly accurate and capable of providing definite dynamical information for the reaction[48].

**Quantum dynamics calculations.** Our quantum dynamics calculations employ the eight-dimensional model originally proposed by Palma and Clary[32] by restricting the non-reacting $CH_3$ group under $C_{3V}$ symmetry. Since the substitution reaction has a saddle point with $D_{3h}$ symmetry, the assumption should hold very well in this study and the model is expected to have a high level of accuracy as having been demonstrated in many studies on the abstraction reaction process[38,39,49,50]. Furthermore, because the length of the non-reacting CH bond essentially does not change during the reaction, it was fixed at the equilibrium value of 2.06 Bohr for $CH_4$, reducing the degrees of freedom included in our calculations to seven. Initial state selected wave packet method was employed to calculate total reaction probabilities for ground and some vibrationally excited initial states as a function of collision energy for some isotope combinations[34]. Transition state wave packet calculations were also carried out to obtain cumulative reaction probabilities from which the thermal rate constants can be evaluated[35,39]. Details of the calculations are provided in Supplementary Information.

**Data availability.** The data that support the findings of this study are available from the corresponding author on request.

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

## Acknowledgements

This work was supported by the National Natural Science Foundation of China (grant nos 21433009, 21688102 and 21403223), the Ministry of Science and Technology of China (2013CB834601) and the Chinese Academy of Sciences (XDB17010000).

## Author contributions

D.H.Z. and S.L. conceived and supervised the research; Z. Zhao, Z. Zhang and S.L. performed the research; Z. Zhao and D.H.Z. analysed the data; and D.H.Z. and S.L. wrote the manuscript.

## Additional information

**Competing financial interests:** The authors declare no competing financial interests.

**How to cite this article**: Zhao, Z. *et al.* Dynamical barrier and isotope effects in the simplest substitution reaction via Walden inversion mechanism. *Nat. Commun.* **8,** 14506 doi: 10.1038/ncomms14506 (2017).

**Publisher's note**: 

