## [Peer Review File · Nature Communications]

REVIEWERS' COMMENTS:

Reviewer #1 (Remarks to the Author):

[Redacted]

Reviewer #2 (Remarks to the Author):

This is an excellent paper on the dynamics of the H + CH₄ substitution reaction and its isotope-substituted analogues. The employed reduced-dimensional quantum model and the ab initio potential energy surface are state-of-the-art providing definitive predictions for the cross sections and reaction probabilities. Furthermore, the qualitative picture based on the analysis of the change of the umbrella angle and the determination of effective dynamical barriers makes this work very significant. Therefore, I strongly recommend publication of this manuscript in Nature Communications.

Minor comments:

The main text suggest that 8D quantum computations were performed: "Our calculation employs the eight-dimensional (8D) model". However, according to the SI, the CH bond length is fixed, thus the computations are actually 7D. This should be clarified in the main text.

On pages 1, 2, and 3: "biomolecular" should be "bimolecular".

On page 4: "augcc-PVTZ" should be "aug-cc-pVTZ".

"CHD₃ reagent of 109.0" Is it a vibrationally averaged value? The equilibrium bond angle in methane is close to 109.5.

The effect of translational energy vs. vibrational excitation is discussed, but the Polanyi rules are not mentioned. The authors may want to add a sentence about the validity of the Polanyi rules, which could add to the significance of the present work.

In Figures 2 and 4: "Bohr" should be "bohr".

In the SI the authors write that "The length of the non-reacting CH bond was fixed at its equilibrium value of 2.06 bohrs." Is it fixed at the equilibrium CH distance of the reactant molecule or at the transition-state value of the CH distance? The latter cannot be called equilibrium, because equilibrium structures correspond to minima.

In the SI the H+CH₃ system is mentioned, but in this work results are only presented for T + CH₄/CD₄.

Reviewer #3 (Remarks to the Author):

The calculations presented in this manuscript are pioneering and state-of-the art. The work will of interest to a broad group of scientists. However, before publication the manuscript needs some revisions as described in the following. The major work that needs to be done is that the written English for the manuscript needs to be vastly improved. I have noted a number of places where changes need to be made, but the manuscript needs to be carefully edited by someone quite proficient in English.

Specific Comments:

1. When it is stated in the manuscript that the reaction threshold found from the dynamics calculation is larger than the barrier height, it should be stated that this is the threshold versus collision energy. It would be good to include this statement on lines 14 and 62, and maybe other places in the text. If TST is valid for the reaction the reaction threshold versus T will be the barrier height. TST would be invalid if there is substantial barrier recrossing.

2. With reference 8 on line 46, it would be good to include *J. Phys. Chem. A* **102**, 6208 (1998), which is a chemical dynamics simulation that reproduces and explains the experimental finding.

3. The experimental determinations of reaction thresholds versus collision energy, for S_N2 reactions, which are larger than the barrier height results from direct reactions. In both experimental studies and chemical dynamics simulations for the exothermic $Cl^- + CH_3Br$ and $Cl^- + CH_3I$ S_N2 reactions, *J. Chem. Phys.* **118**, 2688 (2003) and **138**, 114309 (2013), the reaction mechanism changes from indirect to direct with increase in collision energy.

4. Grammatical corrections:

Line 23 - "...in the gas-phase..."

Line 51 - "intramolecular"

Line 57 - "...effects on the kinetics and..."

Line 59 - change "unfolded" to "unclear"

Line 92 - change "semiempirical" to "model analytic"

Line 99 - change "practicable" to "practical"

Line 167 - ".....responding to the attack..."

Line 211 - "...faster to respond..."

Line 217 - ".....would enhance the efficacy of translational energy on promoting reaction."

Line 244 - ".....looks intriguing because..."

Line 245 - "From the initial state selected..."

5. In the discussion of the difference between theory and experiment in line 267, the authors might comment on the possibility that constraining the reactive system to C_{3v} and using an incomplete representation of rotation in the theoretical calculations might affect the calculated cross section.

First of all, I would like to express my sincere gratitude to all the referees for their positive and constructive comments/suggestions to improve our manuscript. We have carefully revised the manuscript accordingly. This letter details our point-to-point response to all the issues raised by the referees. Hope you will be satisfied with our correspondences.

Point-to-point response to all the issues raised by the referees:

Reviewer #1 (Remarks to the Author):

[Redacted]

[Redacted]

Reviewer #2 (Remarks to the Author):

This is an excellent paper on the dynamics of the H + CH₄ substitution reaction and its isotope-substituted analogues. The employed reduced-dimensional quantum model and the ab initio potential energy surface are state-of-the-art providing definitive predictions for the cross sections and reaction probabilities. Furthermore, the qualitative picture based on the analysis of the change of the umbrella angle and the determination of effective dynamical barriers makes this work very significant. Therefore, I strongly recommend publication of this manuscript in Nature Communications.

Minor comments:

The main text suggest that 8D quantum computations were performed: "Our calculation employs the eight-dimensional (8D) model". However, according to the SI, the CH bond length is fixed, thus the computations are actually 7D. This should be clarified in the main text.

Indeed, we only included seven degrees of freedom in our calculation by fixing the CH bond length. We clarified this in Method part on lines 351-353.

On pages 1, 2, and 3: "biomolecular" should be "bimolecular".

On page 4: "augcc-PVTZ" should be "aug-cc-pVTZ".

Fixed.

"CHD₃ reagent of 109.0" Is it a vibrationally averaged value? The equilibrium bond angle in methane is close to 109.5.

Yes, the equilibrium bond angle is 109.47° , and the vibrationally averaged value is 109.36. We changed it to 109.36 on line 169.

The effect of translational energy vs. vibrational excitation is discussed, but the Polanyi rules are not mentioned. The authors may want to add a sentence about the validity of the Polanyi rules, which could add to the significance of the present work.

We added one sentence on lines 225-227 about the validity of Polanyi's rules.

In Figures 2 and 4: "Bohr" should be "bohr".

Fixed.

In the SI the authors write that "The length of the non-reacting CH bond was fixed at its equilibrium value of 2.06 bohrs." Is it fixed at the equilibrium CH distance of the reactant molecule or at the transition-state value of the CH distance? The latter cannot be called equilibrium, because equilibrium structures correspond to minima.

The CH bond length is fixed at the equilibrium value of the reactant molecule, because it essentially does not change during the reaction on lines 351-353.

In the SI the H+CHT3 system is mentioned, but in this work results are only presented for T + CH4/CD4.

The content for the H+CHT3 system has been removed in SI.

Reviewer #3 (Remarks to the Author):

The calculations presented in this manuscript are pioneering and state-of-the art. The work will of interest to a broad group of scientists. However, before publication the manuscript needs some revisions as described in the following. The major work that needs to be done is that the written English for the manuscript needs to be vastly improved. I have noted a number of places where changes need to be made, but the manuscript needs to be carefully edited by someone quite proficient in English.

Specific Comments:

1. When it is stated in the manuscript that the reaction threshold found from the dynamics calculation is larger than the barrier height, it should be stated that this is the threshold versus collision energy. It would be good to include this statement on lines 14 and 62, and maybe other places in the text. If TST is valid for the reaction the reaction threshold versus T will be the barrier height. TST would be invalid if there is substantial barrier recrossing.

We added “versus collision energy” after threshold in a number of place in the text. See line 17, 51, 135, 206.

2. With reference 8 on line 46, it would be good to include J. Phys. Chem. A 102, 6208 (1998), which is a chemical dynamics simulation that reproduces and explains the experimental finding.

Added in as Ref. 14.

3. The experimental determinations of reaction thresholds versus collision energy, for SN2 reactions, which are larger than the barrier height results from direct reactions. In both experimental studies and chemical dynamics simulations for the exothermic Cl⁻ + CH₃Br and Cl⁻ + CH₃I SN2 reactions, J. Chem. Phys. 118, 2688 (2003) and 138, 114309 (2013), the reaction mechanism changes from indirect to direct with increase in collision energy.

We added in these two references as Ref. 15 and 17, and added a sentence on lines 58-60 regarding the direct reaction mechanism.

4. Grammatical corrections:

Line 23 - “...in the gas-phase...”

Line 51 - “intramolecular”

Line 57 - “....effects on the kinetics and....”

Line 59 - change “unfolded” to “unclear”

Line 92 - change “semiempirical” to “model analytic”

Line 99 - change “practicable” to “practical”

Line 167 - “.....responding to the attack...”

Line 211 - “....faster to respond...”

Line 217 - “.....would enhance the efficacy of translational energy on promoting reaction.”

Line 244 - “.....looks intriguing because...”

Line 245 - “From the initial state selected....”

Fixed.

5. In the discussion of the difference between theory and experiment in line 267, the authors might comment on the possibility that constraining the reactive system to C3V and using an incomplete representation of rotation in the theoretical calculations might affect the calculated cross section.

We added a few sentences on lines 300-309 to discuss these possible error sources for theory.

----- END -----